



# A new method to correct the ECC ozone sonde time response and its implications for "background current" and pump efficiency

Holger Vömel[1], Herman G. J. Smit[2], David Tarasick[3], Bryan Johnson[4], Samuel J. Oltmans[4],
Henry Selkirk[5], Anne M. Thompson[6], Ryan M. Stauffer[6], Jacquelyn C. Witte[1], Jonathan Davies[3],
Roeland van Malderen[7], Gary A. Morris[8], Tatsumi Nakano[9], Rene Stübi[10]

[1]National Center for Atmospheric Research, Boulder, CO, 30301, USA
[2]Research Center Jülich, Germany
[3]Environment Canada, Downsview, ON, Canada
[4]National Oceanic and Atmospheric Administration, Boulder, CO, 30305, USA
[5]University Space Research Associates, Greenbelt, Maryland, USA
[6]NASA Goddard Space Flight Center, Greenbelt, Maryland, USA
[7]Royal Meteorological Institute, Brussels, Belgium
[8]St. Edward's University, Austin, TX, USA
[9]Japan Meteorological Agency, Tokyo, Japan
[10]MeteoSwiss Aerological Station, Payerne, Switzerland

*Correspondence to*: Holger Vömel (Voemel@ucar.edu)

**Abstract.** The Electrochemical Concentration Cell (ECC) ozonesonde has been the main instrument for in situ profiling of ozone worldwide; yet, some details of its operation, which contribute to the ozone uncertainty budget, are not well understood. Here, we investigate the time response of the chemical
reactions inside the ECC and how corrections can be used to remove some systematic biases. The analysis is based on the understanding that two reaction pathways involving ozone occur inside the ECC that generate electrical currents on two very different time scales. A slow reaction pathway involving the buffer with a time constant of about 25 min can be interpreted as what has conventionally been considered the "background current". This contribution can be calculated and removed from the measured current instead
of the "background current". The remaining fast reaction pathway with a time constant of about 20 s is due the conversion of iodide to molecular iodine and the generation of two free electrons per ozone molecule. Here we provide an algorithm to calculate and remove the contribution of the slow reaction pathway and to correct for the time lag of the faster reaction pathway.

This processing algorithm has been applied to ozonesonde profiles at Costa Rica and during the Central
Equatorial Pacific Experiment (CEPEX) and to laboratory experiments evaluating the performance of ECC ozonesondes. At Costa Rica, where a 1% KI, 1/10th buffer solution is used, there is no change in the derived total ozone column; however, in the upper troposphere and lower stratosphere, average reported ozone concentrations increase by up to 7% and above 30 km decrease by up to 7%. During CEPEX, where a 1% KI, full buffer solution was used, ozone concentrations are increased mostly in the upper troposphere with
no change near to the top of the profile. In the laboratory measurements, the processing algorithms have been applied to measurements using all current sensing solutions and using only the stronger pump





efficiency correction reported by Johnson et al. (2002), which improves the time response of the ECCs and removes some biases relative to the reference instruments.

In the surface layer, the correction algorithm shows that ECC ozonesonde measurements are influenced by the operational procedures prior to launch and that typical gradients above the surface layer may be steeper than originally reported.

## Introduction

The Electrochemical Concentration Cell (ECC) ozonesonde is one of the most important instruments for the measurement of vertical profiles of ozone and is used in a number of important networks, e.g. the
ozonesonde network of Global Atmosphere Watch (GAW), the Southern Hemispheric ADditional OZonesondes (SHADOZ) and the Network for the Detection of Atmospheric Composition Change (NDACC). It provides observations of high fidelity and high vertical resolution, which among others are considered a reference for satellite based remote sensing observations. Its operation has been described in detail elsewhere (e.g. Komhyr, 1969; Kohmyr and Harris, 1971; Smit et al., 2014; Sterling et al., 2018;
Tarasick et al., 2020).

The ECC generates an electrical current through the reaction of ozone in a potassium iodide (KI) solution, which produces approximately two electrons per molecule of ozone. The ozone partial pressure ($P_{O_3}$) is then calculated using the ECC equation:

$$P_{O_3} = c \cdot T \cdot t_{100} \cdot \gamma \cdot I_{O_3} \tag{1}$$

with

$$I_{O_3} = I_m - I_B \tag{2}$$

where $P_{O_3}$ is in [mPa]; $I_{O_3}$ in [µA] is the cell current attributed to the reaction of ozone with iodide; $I_m$ in [µA] is the measured cell current; $I_B$ has historically been called the "background current" generated in the absence of ozone; c = 4.309·10$^{-4}$ is the ratio of ideal gas constant and Faraday constant divided by the yield ratio of 2 electrons per ozone molecule; T in [K] is the air temperature entering the cell, approximated by the temperature of the pump; $t_{100}$ in [s] is the flow rate time to pump 100 ml; and γ is a pressure dependent
pump flow correction factor. Other efficiency corrections may be included (e.g. Witte et al., 2017; Sterling et al. 2018; Tarasick et al., 2020), but are omitted here for simplicity.

Prior to launch on a meteorological sounding balloon, ECC ozonesondes are prepared largely following standard operating procedures, which are described in GAW report 201 (Smit and ASOPOS panel, 2014)





and which are currently under review. A central step during the preparation of the ECC is the exposure of

the cell to defined amounts of ozone, typically for 5 min. The amount of ozone is regulated such that the cell generates an electrical current of 5 µA. After ozone exposure, air free of ozone is pumped through the cell and the decay of the cell current is measured. Typical parameters measured are the time during which the cell current drops from 4 µA to 1.5 µA (about 20 s) and the cell current 10 min after exposure to ozone has ended. In addition, the time the pump takes to sample 100 ml air is measured.

The "background current" $I_B$ used in Equation (2) has been assumed to be the cell in the absence of ozone and is a major contribution to the uncertainty of ozone measurements, particularly, in the tropical upper troposphere and in the boundary layer of clean regions of our atmosphere, where ozone concentrations are low (Witte et al., 2018, Tarasick et al., 2020). $I_B$ is treated as a constant offset from the measured current throughout the profile and is measured multiple times as part of the standard operating procedures; however,

there is inconsistency about which of these measurements should be used as the final $I_B$ in Equation (1). In current data records, $I_B$ may have been taken as the cell current prior to the conditioning of the cell with ozone (IB0), as the cell current 10 min after conditioning (IB1), as the cell current using an ozone destruction filter just before launch (IB2), or as a constant value used for all sondes. A decaying background, recommended by one sonde manufacturer (SPC, 2014), is less well defined and has caused additional

ambiguity in processing and interpreting of ozonesonde observations. The arbitrary nature of this term introduces uncertainty that is difficult to quantify. Here, we investigate how the temporal response of the ECC controls the "background current" and how this may be used to improve the processing of ECC ozonesonde measurements.

Vömel and Diaz (2010, hereafter VD2010) studied the cell current during preparation of the ECC in more

detail and pointed out that the concept of a constant background is not supported by the behavior of the instrument during preparation. After exposure to ozone, the measured cell current continues to decrease with a slow time constant of about 25 min. Although the absolute value of the cell current during this decrease differs between the three different solutions, the slow rate of decay of the cell current after ozone exposure is similar for these three solution types. In none of their tests was a constant level established that

could be justifiably used as constant background in the calculation of the ozone partial pressure.

Throughout the ECC ozonesonde community, these instruments are operated using predominantly three chemical solution recipes; these differ mostly in the relative strength of the potassium iodide and the strength of the buffer [see Johnson et al. (2002) and VD2010]. The original solution recipe introduced by Komhyr (1986) will be referred to as the 1% KI, full buffer solution and has been used in many ozone

soundings including those during the Central Equatorial Pacific Experiment (CEPEX; Kley et al., 1996 and VD2010). When it was understood that the buffer in the solutions not only regulate the pH value but also





contribute to the generation of excess electrons, Komhyr (EnSci, 1996) proposed to dilute this original recipe by a factor of two. This recipe will be referred to as the 0.5% KI, ½ buffer solution. Sterling et al. (2018) introduced a third solution, in which only the strength of the buffer was reduced by a factor of 10

while maintaining the original concentration of potassium iodide. This solution will be referred to as 1% KI, 1/10th buffer solution and is has been used across the NOAA ozonesonde network as well as in Costa Rica.

VD2010 also pointed out that for many field stations the availability of ozone free air is limited. Purified air using ozone destruction filters are most commonly used at both operational and campaign driven sites.

The assumption that these filters operate with perfect efficiency and reliance under all conditions cannot be assumed (Reid et al., 1996; Newton et al., 2016; Witte et al., 2017). Therefore, the measurement of the cell current after the exposure to ozone using such filters may still include some contribution from the reaction of residual ozone and iodide, further complicating the determination of a "background current".

Here, we argue, that the term "background current" is a misnomer and suggest that the term "post-

preparation current" is more suitable, tying this term to the standard operating procedures and referring explicitly to the cell current measurement 10 min after the exposure of ozone. This preparation current provides valuable information about the functionality of the sensor and connects to the established record of ECC operations over the past 50 years.

VD2010 emphasized the role of side reactions of the buffer with ozone and the time dependence of the

different reaction pathways, which may generate electrical currents in excess of the conversion efficiency of two. Tarasick (2020) proposed considering the different reaction pathways explicitly and deriving a quantitative method linking the slow side reactions to what has historically been called the "background current". Here, we explore this proposal further and evaluate a quantitative algorithm, which takes into account the slow reaction path involving the buffer as well as a correction for the time response delay of

the fast reaction path in the reaction between ozone and iodide. We argue that the preparation current should not be used in the calculation of the ozone partial pressure and that its role is replaced by the explicitly calculated contribution of the slow reaction path.

Lastly, the pump flow correction factor compensates for a reduced pump efficiency at low pressure, which becomes relevant at pressure less than 100 hPa, i.e. in the stratosphere. Three pump flow correction tables

are currently in widespread use (Komhyr, 1986; Komhyr et al., 1995; and Johnson et al., 2002; see Smit and ASOPOS panel (2014) for more detail), which in the middle stratosphere (5 hPa) differ by as much as 15%. The pump flow corrections by Komhyr (1986) and Komhyr et al. (1995) and are recommended for sondes using the more strongly buffered solutions (1% KI, full buffer, and 0.5% KI, ½ buffer respectively). The pump flow correction by Johnson et al. (2002), which provides a stronger correction than the other





two, is recommended only for sondes using the 1% KI, 1/10th buffer solution. By pairing these recommendations, systematic biases due to the generation of excess electrons in a particular sensing solution are compensated by the matching pump efficiency correction. However, only the pump flow correction by Johnson et al. (2002) is currently recognized as properly describing the loss of pump efficiency and consistent with measurements from other groups (Tatsumi Nakano, personal

communication). Pairing this pump efficiency with the more strongly buffered solutions would lead to an overestimation of stratospheric ozone. Here we argue that this inconsistency can be addressed by explicitly calculating the contribution of the slow reaction path in the stratosphere, which properly compensates for the generation of excess electrons by the more strongly buffered solutions.

In the calculation of the total ozone column, we use the satellite climatology by Peters and Labow (2012)

to estimate the amount of ozone not measured by the ECC above the balloon burst or above 7 hPa, whichever comes first. Using this climatology and the limit to 7 hPa for the top of the ozone sonde profile reduces the influence of the strongest pump efficiency correction near the top of the profile.

**Method**

VD2010 presented an equation to describe the decay of the ECC cell current after the exposure of ozone in

the laboratory:

$$I(t) = I_0 e^{-\frac{t}{\tau}} + I_0' e^{-\frac{t}{\tau'}} \qquad (3)$$

where $\tau \approx 20$ s and $\tau' \approx 25$ min. This equation contains a fast term, which is due to the reaction of ozone with potassium iodide with a value of $I_0$ at $t$=0 and a slow term, which is due to additional side reactions involving the buffer with a value of $I_0'$ at $t$=0. The two values $I_0$ and $I_0'$ at $t$=0 are unknown and cannot be determined with a single measurement at $t$=10 min.

Komhyr (1969) and Komhyr and Harris (1971) attribute the fast time constant to diffusive transport of iodine through the diffusion layer to the cathode electrode. They report a time constant of faster than 20 s at 25°C with a strong temperature dependence and a slowing to 40 s at 2°C. Saltzman and Gilbert (1959) and Flamm (1977) attribute the slow reaction path to additional reactions involving the neutral phosphate buffer used in the sensing solutions. Flamm (1977) determined at time constant of 27.4 min, Tarasick et al.

(2020) use a time constant of 20 min for the slow reaction path.

The decay of the cell current signal differs in magnitude between the two solution recipes studied by VD2010, even though the time constants for the two solution recipes are very similar. Therefore, we concluded along with others (e.g. Johnson et al., 2002) that the concentration of the buffer is the main cause



for the different responses. This also implies that the different solution recipes may be handled
mathematically in the same way, only differing in some parameters.

With the assumption of symmetric reaction kinetics, we can estimate the contribution of each of the two
reactions separately. We may therefore interpret Equation (1) to mean that the measured cell current has a
contribution $I_0$ coming from the reaction of ozone with iodide and a contribution $I_0'$ coming from the
unspecified secondary pathway. Only the current $I_0$ generated in the fast primary reaction with iodide
should be used in the calculation of the ozone partial pressure. In contrast, the current contribution $I_0'$
generated from the secondary reactions must be considered as an excess current that should be subtracted
from the measured cell current. Therefore, the term, which has in the past been considered a constant
"background current", should rather be considered a time-dependent excess current due to the secondary
reactions within the ECC.

Equation (3) may be generalized as

$$\frac{dI_m}{dt} = -\frac{1}{\tau}(I_o - I_{ss}) - \frac{1}{\tau'}(I_0' - I_{ss}') \tag{4}$$

where $I_m = I_0 + I_0'$ is the measured cell current, $I_0$ is the instantaneous contribution of the fast reaction, $I_0'$
is the instantaneous contribution of the slow reaction, $I_{ss}$ is the steady state contribution of the fast reaction,
and $I_{ss}'$ is the steady state contribution of the slow reaction.

To understand the partition between the two reaction pathways, we assume that for long time scales the fast
reaction is in near steady state, which allows integrating the slow reaction pathway separately. Thus, we
can write the time dependence of the slow reaction pathway as

$$I_0'(t) = \left(1 - e^{-\frac{t-t_0}{\tau'}}\right) \cdot I_{ss}' + e^{-\frac{t-t_0}{\tau'}} \cdot I_0'(t_0), \tag{5}$$

which can be evaluated iteratively over all time steps of a profile beginning with the start of data recording
prior to launch. This equation assumes some knowledge of the slow pathway contribution $I_0'(t_0)$. If an
ozone destruction filter was used as part of the launch preparation procedures, then the cell current reading
at $t_0 = 0$, i.e. the moment just before the filter was removed (IB2), is equivalent to the slow reaction
pathway only. Without the use of an ozone destruction filter, the slow pathway contribution $I_0'(t_0)$ must
be assumed. The contribution of the slow reaction pathway at the surface $I_0'(t_0)$ decreases exponentially
as the sounding progresses, which may justify abandoning the concept that IB2 or any other arbitrary value
should be applied as constant background throughout the profile.



After removing the ozone destruction filter before launch or after the conclusion of the ECC preparation, the measured cell current becomes the superposition of both pathways. The uncertainty in the development of the slow pathway prior to launch is the largest contribution to the uncertainty of the measurements in the boundary layer, but decreases as the contribution of the pre-launch reading decreases. Variations in operational procedures, such as when is the ozone destruction filter removed and time elapses between the end of the ozone conditioning and launch, contribute to the uncertainty.

At the same time, the contribution of the slow pathway steady state value $I'_{ss}$ increases exponentially. This steady state value cannot be measured directly during a sounding, but has been determined in laboratory experiments. VD2010 measured an excess cell current as a function of cell current under steady state conditions for the three solution recipes and determined a linear relationship between the excess and cell current (their Figure 4). This measurement shows the steady state contribution of the slow reaction pathway, which is directly proportional to the measured cell current. This can be used to write Equation (5) as

$$I'_0(t) = \left(1 - e^{-\frac{t-t_0}{\tau}}\right) \cdot \alpha \cdot I_m + e^{-\frac{t-t_0}{\tau'}} \cdot I'(t_0) \tag{6}$$

where the steady state $I'_{ss}$ has been replaced by the measured cell current multiplied by a steady state bias $\alpha$. VD2010 derived steady state bias factors of $\alpha$=0.090±0.005 for the 1% KI, full buffer solution, $\alpha$=0.031±0.004 for the 1% KI, 1/10[th] buffer solution, and $\alpha$=0.024±0.009 for the 0.5% KI, ½ buffer solution.

Equation (6) allows the calculation of the contribution of the slow reaction pathway based on the time constant $\tau' \approx 25$ min, the measured cell current, and an assumed or measured slow reaction pathway cell current $I'_0(t_0 = 0)$ prior to launch. Integration preferably starts with pre-launch measurements, but in practice may be limited to calculations starting at launch. In that case, the uncertainty of the initial slow reaction pathway may be significant, depending on the total amount of ozone in the near surface boundary layer.

With the knowledge of the slow reaction pathway, we can evaluate the time response of the fast reaction pathway and remove the time lag, which is introduced by the response time of about 20 s. Removing the contribution of the slow reaction to separate out the fast reaction contribution, we can write

$$I_0(t) = I_m(t) - I'_0(t) \tag{7}$$

and

$$\frac{dI_0(t)}{dt} = -\frac{1}{\tau}(I_o - I_{ss}) \tag{8}$$



Here, $I_0(t)$ is the contribution of the fast reaction to the measured cell current and $I_{ss}$ is the instantaneous steady state value, i.e. the fast reaction cell current that would be measured if the ozone concentration was in steady state. Equation (8) can be rearranged to

$$I_{ss} = I_o + \tau \cdot \frac{dI_0}{dt} \qquad (9)$$

and can be used to calculate the time lag corrected ozone concentration. This equation is identical to the equation derived by Huang et al. (2015) and removes a small bias in the fast reaction pathway due to its still considerable time constant of approximately 20 s. This steady state cell current reflects only the fast reaction pathway due to the reaction of ozone with iodide and represents the current $I_{O_3}$ that is used in Equation (1) to calculate the ozone partial pressure.

Any level of noise in the raw data will be amplified by the term $\tau \cdot \frac{dI_0}{dt}$ introducing an additional random uncertainty, which is proportional to the time constant and the ozone gradient. Here, we smooth the fast component $I_0(t)$ of the cell current with a Gaussian filter using a width equal to 20% of the time lag constant.

$$I_{0,smooth}(t) = \frac{1}{k} \sum_{-t_i}^{+t_i} e^{-\frac{t_i^2}{2\sigma^2}} \cdot I_0(t + t_i) \qquad (10)$$

with

$$k = \sum_{-t_i}^{+t_i} e^{-\frac{t_i^2}{2\sigma^2}} \qquad (11)$$

where $\sigma = 0.75 \cdot \tau$ and $t_i$ are time steps around the current time $t$. To reduce the computational effort, it is sufficient to use data in the time series of $\pm 3\sigma$ around the current time step for the smoothing.

Using a running mean of equal width works as well, but may produce slightly larger noise and less realistic small structures in the final profile. Other smoothing filters such as B-splines may also be used to reduce noise in the raw data.

To show the effect of removing the slow pathway (Equation (6) and applying the time lag correction (Equation (9), we apply these algorithms to the laboratory measurements of VD2010. Figure 1 shows the measurements of Figure 3 in VD2010, calculated as mixing ratio. This laboratory experiment used the 1% KI, full buffer solution type and sonde 2Z4773. The conventionally derived mixing ratio is shown in





orange, the time response corrected mixing ratio in red. The calculated contribution of the slow pathway is
shown in purple and demonstrates the effect of the slow increase of that pathway. The original
measurements focused on the steady state towards the end of each plateau to avoid the slow reaction path.
In the corrected data shown here, the contribution of the slow reaction has been explicitly removed and
shows a much better agreement with the ozone calibrator. In particular, the slow behavior at the change of
the plateaus has been removed.

The classical processing of the ECC ozonesondes in Equations (1) and (2) assumes a constant "background
current"; however, the contribution of the slow reaction pathway to the measured cell current is anything
but constant. This result shows that using a "constant background" is not valid, regardless of which value
is chosen.

The difference between the corrected ECC mixing ratio and the TEI 49C ozone calibrator (Figure 1, bottom)
is nearly constant with a value of 0.53 ppb, covering the first four step changes over the series and the
pattern differs significantly from the time dependent difference shown in Figure 3 of VD2010. The behavior
of the difference changes after about 5.5 hours, most likely due to evaporation of sensing solution.

The effect of the time lag correction on the response of the ECC during the step changes is shown in Figure
2. These experiments used two different 2Z series ECC sondes from EnSci and one 6A series ECC sonde
from Science Pump Inc. and the three most common sensing-solution recipes. While the originally
processed measurements show the effect of response time lag, the corrected data show a response that is
nearly indistinguishable from the drop of ozone generated by the TEI 49C. In particular, the small bias of
the ECC remains almost constant across any step change.

The measurements show that the time response is nearly identical for these three sondes and sensing
solutions, suggesting that this approach can be applied to the most commonly used sonde types and
solutions. The time lag corrections for the six step changes shown in Figure 2 represent a total of 60 step
changes in 25 different experiments. The correction approach may be applied to any of these instruments
and solutions and could be used at operational stations to remove the effect of the slow reaction in existing
time series. The small biases between the corrected ozone mixing ratio and the TEI 49C may in parts be
due to the accuracy of the TEI 49C calibrator and in parts be specific to the individual sondes or sensing
solutions used in these tests. The small, observed differences may already be representative for the ECC
model or sensing solution type; however, more work would be required to better explain these small
differences.





### Validation in independent laboratory experiments: JOSIE

The World Calibration Center for Ozone Sondes (WCCOS) at the Research Center Jülich has conducted a series of ozonesondes tests, comparing instruments operated by staff from different ozonesonde stations against a reference ozone photometer (OPM). The sondes were tested in the Environmental Simulation Chamber at Jülich, in which temperature, pressure, and ozone mixing ratio were regulated simultaneously to represent a mid-latitude, a sub-tropical, and a tropical profile. Here, we use data from two ozonesondes

tested during the Jülich OzoneSonde Intercomparison Experiment (JOSIE) in September 2000 (Smit et al., 2007).

The two sondes shown here used the 0.5% KI, ½ buffer solution and were originally processed with the pump efficiency correction of Komhyr (1986). We have reprocessed these measurements using the algorithms described above and summarized here: The slow reaction contribution to the measured cell

current was calculated using a time constant of $\tau' \approx 25$ min and a steady state bias of 0.024 (2.4%), based on VD2010. To initialize the calculation, we assumed that the "background" was measured 20 min before the start of the simulation and that the ozonesondes were measuring at the simulated surface value for that period. This calculated slow reaction contribution was subtracted from the raw current instead of any "constant background". The resulting fast reaction cell current was then smoothed using the Gaussian filter

in Equation (10) and corrected for time lag of the fast reaction contribution using the fast time constant reported for each JOSIE simulation, which had been measured prior to each simulation run (on the order of 20 s). In the calculation of the partial pressure and mixing ratio we used the average pump-efficiency correction reported by Johnson et al. (2002) for Science Pump 6A sondes.

Figure 3 shows simulations of a tropical and a mid-latitude profile, including two periods each during which

the ozone concentrations in the chamber was switched to zero to study the time response of the ozonesondes. The original ozonesonde measurements, the reprocessed data, and the difference to the OPM are shown. The pressure approximately followed a typical balloon ascent and is shown as well.

The reprocessing shows some interesting differences. The reprocessed tropical measurements between 55 min and 100 min show on average about 5% higher ozone than the reference, while the original data start

with a low bias of about 10% and then show agreement with the reference. During this time, the reprocessed, (time response corrected) data follow the OPM data slightly better than the originally uncorrected data. At the lowest pressures between 100 min and 120 min, the reprocessed data do not fall off as rapidly as the originally processed data and show good agreement with the reference, while the originally processed data drop to a 10 % low bias. The different pump efficiency correction used in the reprocessing, which corrects

the pump inefficiency at low pressures more strongly, contributes most to this difference, with a smaller contribution by the slow reaction.



In the simulated tropical profile, the reprocessed ECC ozone concentration in the simulated upper troposphere between 30 min and 60 min is larger than the reference, and much larger than the near zero ozone concentrations reported by the original processing. However, since the true ozone concentration is very low, overall uncertainties and relative differences are large in this segment of the profile.

The reprocessed mid-latitude simulation shows only small changes, except at the lowest pressures after about 100 min. Again, the reprocessed data do not drop off as quickly due to the different pump efficiency correction used with the reprocessed data.

Figure 4 shows the different cell current contributions of the original and reprocessed measurements. These data are shown on logarithmic scale to highlight both slow and fast reaction contribution on the same plot. Most importantly, the slow reaction contribution to the cell current may vary by almost a factor of ten in both the tropical and the mid-latitude simulation. This is in contrast to the assumption of a constant "background" in the original processing. The effect is most noticeable in the tropical simulation, where the "background" corrected cell current is much smaller than the fast reaction contribution to the cell current, leading to a strong underestimation of ozone. Past the ozone peak after around 60 to 90 min, the slow reaction contribution is larger than the constant "background" assumption and slightly lowers the calculated ozone. However, since the total ozone concentration is large, the net effect is small. Near the end of the simulation, i.e. at the lowest pressure, the slow reaction contributions become slightly larger and reduces the effect of the larger pump efficiency correction.

There is some uncertainty in the contribution of the slow reaction pathway at the beginning of the simulation, since the history of the ECC chemistry prior to the start of the data recording is not known. Changing the time when the "background" was measured (we assumed 20 min prior to the start of the simulation) has some influence on the slow reaction contribution in the early phase of the simulation.

In both the tropical and mid-latitude simulation, the reprocessed data show a dramatically improved response relative to the OPM reference compared to the originally processed data. The zero ozone periods in both the tropical and mid-latitude simulation after about 60 min are shown in Figure 5 and demonstrate that the reprocessed ozone partial pressures closely follows those of the OPM. Results are very similar to the earlier zero ozone periods at 15 min, confirming the improvement already seen in the lab measurements shown in Figure 2.

The integrated ozone amount in the reprocessed profiles is about 5% larger than the OPM integrated ozone for both the tropical and mid-latitude simulation. This is slightly worse compared the original processing, which had shown agreement with the OPM in the tropical simulation and a 3% larger value for the mid-latitude simulation. However, in the reprocessed simulations, the excess is almost constant throughout the





entire profile, in contrast to compensation of excess and shortage in the original processing. These
remaining biases indicate that not all sources of uncertainty have been captured yet; however, the
improvement in consistency indicates a better understanding of the role played by the slow reaction
contribution.

The JOSIE 2017 campaign tested over 70 different sondes with a combination of sensing solutions and
sonde manufacturers. Preliminary results are shown by Thompson et al. (2019) and these data are currently
analyzed in more detail. Here, we applied the time response corrections to all simulations using the steady
state bias matching the respective sensing solution, the fast time response provided with each sonde run,
and a slow time constant of 25 min. Furthermore, all simulations are processed using the pump efficiency
by Johnson et al. (2002). The average difference between the original and corrected sonde data and the
OPM data is shown in Figure 6. While there are many details in this data set, which are smoothed out by the
averaging and require a more detailed analysis, we show that the structure in the difference profile is
strongly reduced and that on average the ECC sondes agree with the OPM to well within 3% throughout
all pressures.

Differences due to sensing solution and manufacturer still require careful analysis; however, the much better
agreement after applying the time response corrections shows that the time behavior of the ECC
ozonesondes must be considered in the analysis of ECC ozonesonde data.

**Application to atmospheric measurements**

We processed the series of ozonesonde observations at Costa Rica using the algorithm introduced above to
evaluate its impact on real world observations. At that site, we have used the 1% KI, 1/10th buffer solution
since the beginning, with the exception of a short period when the 0.5%, ½ buffer solution was used. All
soundings with the 1% KI, 1/10th buffer solution used the pump efficiency correction by Johnson et al.
(2002), while the sounding with the 0.5%, ½ buffer solution used the pump efficiency correction by Komhyr
(1995). As described above, the steady state bias for the 1% KI, 1/10th solution was assumed to be 3.1%
and that for the 0.5%, ½ buffer solution was assumed to be 2.4% based on the measurements by VD2010.
Same as for the JOSIE simulations, the contribution of the slow reaction pathway was calculated and
subtracted from the measured cell current. The remaining fast reaction contribution was smoothed and
corrected for time lag. Figure 7 shows a profile measured at Heredia, Costa Rica, in 2010 and its reprocessed
profile. The largest difference is in the upper troposphere and lowermost stratosphere with over 20 % larger
reported ozone concentrations after reprocessing. At the top of the profile, the reprocessed ozone
concentration is 5% lower.





The contribution of both the slow and the fast reaction pathway is shown in Figure 8. The constant
       "background current" used in the original processing is shown for reference. Up to about 23 km, the cell
       current contribution of the slow reaction is less than what was assumed with the original "background
       current". In the stratosphere, the contribution of the slow reaction pathway exceeds the original
       "background current" due to the slow buildup of secondary reaction products. This implies that the ozone
profile in the troposphere has been slightly underestimated and slightly overestimated in the stratosphere.

       The effects of smoothing and correcting for time lag are shown in Figure 9, where we show a close-up of
       the profile shown in Figure 7. This particular sounding exhibits two significant peaks between 11 km and
       14 km. The originally analyzed profile is shown in blue. The first processing step (orange) removes the
       contribution of the slow reaction pathway, followed by the Gaussian smoothing (purple), and finally the
time lag corrected profile (red) on the right of the set of profiles. The difference between constant
       "background" subtraction and removing the slow reaction component is evident in the agreement between
       the original and corrected profile at 10 km and a difference of about 15 % at 15 km.

       The time lag correction enhances both peaks by about 5% and places them at a lower altitude than the
       uncorrected measurements by about 100 m to 150 m. The amplification of features depends on the vertical
gradient [see Equation (9)]. In this example, the lower peak at 12 km is amplified stronger than the upper
       peak at 13.3 km because of its steeper gradient at a nearly identical rise rate of the balloon.

       The noise amplitude of the time lag corrected data is comparable to that of the original data, but its spectral
       characteristics are different as a result of the smoothing algorithm. Therefore, scientific analyses should be
       based on layer averages, since individual data points are heavily influenced by the noise characteristic of
the smoothed data.

       The behavior of the same ozone profile at and shortly after launch is shown in Figure 10. The gradient of
       ozone above the surface layer is strongly enhanced by the time lag correction and appears even stronger in
       the corrected data than in the uncorrected data. (Note, that in the laboratory experiments shown above, even
       stronger gradients are well represented after the corrections have been applied.) Furthermore, the measured
ozone mixing ratio at launch depends on the history of the ECC prior to launch and therefore the operational
       procedures prior to launch. In Figure 10, we show two profiles with different assumptions on the pre-launch
       history of the ECC. The purple trace assumes that the 5 µA ozone conditioning was stopped 40 min prior
       to launch and that the ECC was then exposed to zero ozone air for 10 min, when it reached a preparation
       cell current reading of 0.05 µA. After that, it is assumed that the sonde was moved to the launch site and
continued measuring until launch. The orange trace assumes that an ozone destruction filter was used
       between the 5 µA ozone conditioning and 5 min prior to launch. The difference between both cases is about
       10% at launch and rapidly decays after launch. The time between ozone conditioning and launch as well as



the time prior to launch during which the ECC was exposed to ambient ozone is highly variable. As a result, the surface reading of operational ECC sondes at launch contain significant uncertainties, which decay

within the first couple of km as the ozone concentration above the surface increases and the influence of the operational procedures decreases.

Newton et al. (2016) reported ozonesonde measurements from the Western Pacific, where, due to failure of their sonde preparation equipment, a number of sondes were launched with very high "background currents", which had to be corrected by an ad-hoc hybrid background correction. We believe that this

problem could be addressed using Equation (6) and an appropriate choice for the slow reaction contribution at the surface.

We have applied the correction algorithm to 577 ozonesondes launched at Costa Rica, which allows us to evaluate its impact statistically. The median difference between the corrected and originally reported ozone profiles is shown in Figure 11. Here, we show the influence of only removing the slow reaction contribution

and of removing the slow reaction contribution and applying the time lag correction.

Three features of the complete correction of removing the slow reaction contribution and applying the time lag correction can be highlighted.

1) The surface layer readings are significantly increased with the new correction algorithm. However, the surface reading itself has a larger uncertainty than the rest of the profile. This effect disappears

approximately 1 – 1.5 km above the surface.

2) The ozone mixing ratio in the upper troposphere and lower stratosphere between 10 km and 25 km is larger as a result of these corrections, with the largest correction at the tropical tropopause. This increase is due to both processing steps, i.e. the smaller contribution of the slow reaction compared to the constant "background current" processing and the time lag correction. In fact, at the tropopause at about 17 km, the

change is mostly due to the time lag correction and less to the smaller slow reaction contribution. The overall increase in this region is due to the mean shape of the tropical ozone partial pressure profile, which has its maximum around 25 km.

3) The ozone mixing ratio near the top of the profile decreases on average by about 5 %, which is in about equal parts due to the removal of the larger slow reaction contribution and the time lag correction. The

influence of the time lag correction is again due to the climatological shape of the tropical ozone profile above the mean ozone partial pressure maximum. This change improves agreement with simultaneous MLS observations, which are lower than the Costa Rica sondes for much of the record (Stauffer et al., 2020).

Interestingly, there is very little change in the middle troposphere between about 3 km and 10 km, where the different removal of the slow reaction contribution compared to the constant "background current" is



compensated by the time lag correction. This may be typical for tropical profiles, but not necessarily for mid and high latitude profiles.

VD2010 had suggested using constant steady state bias correction and a fixed small constant "background current" offset without consideration of the temporal response in processing of ozonesondes. The results shown here indicate that the time response of both the fast and slow reaction must be considered and may

have equal contributions to the overall deviations from the simple ECC equation. A simple bias correction as suggested by VD2010 is not sufficient.

Figure 11 indicates that the areas of increased and decreased ozone mixing ratio are approximately equal. For the calculation of the total ozone column, these areas may cancel and the influence on the total ozone column is likely small. Figure 12 shows a histogram of the change in total ozone column for all ozone

profiles at Costa Rica and demonstrates that there is no change at all. The median change is $0.0 \pm 1.1$ DU. Therefore, even though the profile structure is changed, comparisons with observations measuring total ozone column would not be affected by these new processing algorithms, at least for sites such as Costa Rica.

We also applied the correction algorithms described above to the ozone data of CEPEX, which had already

been reprocessed by VD2010 to study the impact of the "background current" on ozone measurements in the upper tropical tropopause. VD2010 argued for a different treatment of the "background current" using a steady state correction approach, in which a modified background depended on the instantaneously measured cell current. In contrast, here we explicitly consider the temporal behavior of the slow and the fast correction pathways separately. Furthermore, we use the pump efficiency correction by Johnson et al.

(2002), instead of the original pump efficiency correction by Komhyr (1986). During CEPEX, the original 1% KI, full buffer solution was used; therefore, we use the steady state bias of 9% in Equation (6) based on the measurements by VD2010.

Figure 13 shows the results explicitly considering temporal characteristics of the slow and fast reaction pathways. The left panel shows the originally processed and the reprocessed CEPEX data. Similar to

VD2010, the most significant effect is in the upper troposphere, which eliminates all of the near-zero ozone observations. The middle panel shows the contribution of the slow reaction contribution in comparison to the original constant "background current" of 0.065 μA and the modified "background" used by VD2010. The slow reaction contribution is similar to the modified background in VD2010 in the upper troposphere, but smaller in the middle and lower troposphere and in the stratosphere, which is due to the slow buildup

of the slow reaction pathway with exposure to ozone. There is a significant spread in the slow reaction contribution near the ceiling of the profile, which is in part also due to the significant variation in the balloon ascent rate during that campaign, giving some sondes more or less time to build up the contribution of the



slow reaction pathway. A simple scaling of the modified background as used by VD2010 overestimates that contribution and slightly underestimates the measured ozone in the stratosphere.

The relative difference of the reprocessed and the original data is shown in the right panel of Figure 13. Similar to VD2010, the largest relative change is in the upper troposphere; however, the less obvious but more important result is that there is virtual agreement between the reprocessed and the original data near the ceiling despite using the stronger pump efficiency correction by Johnson et al. (2002). The mean total ozone column for the CEPEX data set changes by about 7 DU or 3%. The increases in the upper

troposphere, where the change in the pump efficiency correction is insignificant, contributes the majority to this change in the column.

In the reprocessed data, the excess cell current of the full buffer solution is explicitly considered by removing the contribution of the slow reaction pathway. This approach no longer requires the compensation of errors when using the weaker pump efficiency correction by Komhyr (1986) and Komhyr et al. (1995)

to compensate the excess cell current of the stronger buffer solutions. Our approach allows processing of soundings with a proper pump efficiency correction and without the need to match the pump efficiency correction to the sensing solution.

The lowest troposphere also shows a significant increase in the reported ozone after the reprocessing. However, this increase depends on the not well-recorded use of the ozone destruction filter prior to launch.

Here, we assumed that the slow reaction contribution has decayed to 0.02 µA 5 min prior to launch based on scanning the available pre-launch data. However, there may be a significant uncertainty in this assumption.

## Discussion

Processing ECC ozone data with an explicit calculation of the slow reaction path and a time lag correction

for the fast reaction path requires knowledge about three coefficients: the slow reaction time constant, the steady state bias, as well as the fast reaction time constant. In addition, an assumption about the partitioning of the measured cell current between slow and fast reaction pathway at the start of the data series is needed; however, this assumption only influences the calculated ozone mixing ratio in the boundary layer.

For the slow reaction pathway, VD2010 reported values of 24 min for the 1% KI, full buffer solution and

28 min for the 1% KI, 1/10th buffer solution, which is comparable to what has been reported by other studies (e.g. Davies et al., 2000). However, the exact value is not well known and no level of confidence has been determined.





The steady state bias depends on the sensing solution and has been reported by VD2010 and a number of other studies (e.g. Johnson et al., 2002; Smith et al., 2007). The measured values vary considerably, which is, in part, due to the laboratory setup and data analysis. Furthermore, the steady state bias may change during a sounding as water evaporates from the solutions, increasing the concentration of its ingredients. A dependence of the steady state bias on the temperature of the solutions may also be possible and has not been well studied.

A fast reaction time constant is typically measured during the preparation of the ECC sonde and has been used in the analyses above. Komhyr (1971) and Komhyr et al. (1995) report a dependence of the fast reaction time constant on temperature, solution volume, and pressure. During JOSIE (Smit 2007), measurements of faster time constants after the completion of simulation runs were attributed to the evaporation of solutions. Therefore, the time constant measured during the preparation of ECC ozonesondes may not exactly represent the time response during a sounding.

To evaluate the uncertainty of the algorithm depending on the uncertainty of these coefficients, we repeat the correction of the ozone profiles at Costa Rica while independently varying the coefficients used in the correction. The slow reaction time constant is varied by a factor of 2 from 12 min to 50 min. The fast reaction time constant is varied by a factor of 1.5 from $0.66 \cdot \tau$ to $1.5 \cdot \tau$, where $\tau$ is the originally measured fast reaction time constant. The steady state bias reported by VD2010 for the 1% KI, 1/10th buffer solution is varied by a factor of 2 from 1.5% to 6%. We estimate that these intervals cover a range, which includes the true value with a 95% probability (2 sigma).

Figure 14 shows the contributions to the uncertainty of the corrections of the ozonesonde record at Costa Rica shown in Figure 11. The single most important source of uncertainty in the corrections is the uncertainty of the steady state bias, which dominates the uncertainty budget in the free troposphere and the middle stratosphere. Only in the lowermost stratosphere and the surface layer, the regions of the strongest gradients in the ozone profile, is the uncertainty of the fast reaction time constant the dominant contribution. These regions are also the regions experiencing the largest correction. The uncertainty of the slow reaction time constant is secondary throughout the entire profile.

Therefore, further studies, such as a detailed analysis of all JOSIE simulations, may focus on a better quantification of the steady state bias of the different sensing solution recipes.

These uncertainties discussed here describe the mean removal of systematic biases due to the time response of ECCs for the entire data set and may justify the uncertainty of ozonesonde profiles in the validation of remote sensing observations. They do not describe the uncertainty of the correction of individual profiles, which depends strongly on the structure of the each profile.



Furthermore, the corrections and uncertainties discussed here apply only to the time response model described above. Other effects, such as response differences of sondes from different manufacturers and pump related effects are not captured by the processes described here. However, the corrections for time response of the ECC need to be considered in properly quantifying other processes influencing the accuracy of ECC ozonesondes.

**Summary**

Two reaction pathways occur in an ECC ozonesonde, each of which generate electrons: the well understood reaction between ozone and iodide, which generates 2 electrons per ozone molecule, and a secondary slow reaction, which generates additional electrons, but which is not well understood. Here we consider explicitly the time constants of both reaction pathways to derive the ozone partial pressure. The contribution of the

slower secondary reactions to the measured cell current is calculated separately and subtracted from the measured cell current. The remaining fast reaction component is then smoothed using a Gaussian filter and corrected for its lag in response. The resulting corrected fast reaction cell current, which is attributed to the ozone-iodide reaction, is finally used to calculate the ozone partial pressure. This approach overcomes the question whether there is a constant or a decaying "background current" and replaces it with the calculation

of the contribution of the slow secondary reaction.

The algorithm considers the steady state bias of the different sensing solution recipes allowing processing any sensing solution independent of the pump efficiency correction. Selecting weaker and inappropriate pump efficiencies to compensate for side reactions in more strongly buffered solutions is no longer required.

The cell current measured during preparation while ozone free air is pumped through the cell, which has

been called "background current", should rather be called "post-preparation current", and should not be used in the calculations. This measurement is an indication of the proper functioning of the sonde and serves as acceptance criterion for the instrument and the preparation procedure as long as a certain threshold is not exceeded. It is not a property of the sonde that remains constant throughout operation.

The time lag correction of the fast reaction pathway enhances vertical features and removes a systematic

bias, which is introduced in regions of strong gradients due to the relatively slow response relative to the balloon ascent rate, i.e. a low-bias in the region below the ozone peak and a high-bias in the region above the ozone peak.

An initial value for the slow cell current contribution is required during the analysis of the profile data. This value may be derived from experience, it may be measured using a high-quality filter prior to launch, or it

may be set to zero. The contribution of this choice decays with the slow time constant and mostly influences



the uncertainty of the ozone concentration in the boundary layer. It has no influence on the ozone measurements above the middle troposphere, in particular in the tropical upper troposphere, where erroneous "background current" values have led to very large uncertainties (e.g. VD2010; Witte et al., 2018). Specific requirements for the operating procedures prior to launch may help reducing the uncertainty

in the boundary layer and will be included in the revised standard operating procedures.

The net effect of this process on the total ozone column derived from ECC sonde launches at Costa Rica is zero. Therefore, this correction does not affect the comparison with remote sensing instrumentation measuring the total ozone column, at least for the 1% KI, 1/10th buffer solution and using the correct pump efficiency correction measured by Johnson et al. (2002); however, it will affect comparisons with other

profiling instruments.

Reprocessing the CEPEX data using this method achieves a similar result in the upper troposphere as VD2010, but improves the ozone calculation in the stratosphere, since it allows replacing the old incorrect pump efficiency correction by Komhyr (1986) with the better pump efficiency correction by Johnson et al. (2002).

More work is required to properly quantify the steady state bias of the different sensing solutions based on high quality laboratory measurements. The theoretical understanding of both chemical pathways needs to be improved, which may lead to a further refinement of the approach demonstrated here. However, it is clear that including the reaction dynamics in the processing already removes some systematic biases, which have previously only been addressed through ad-hoc methods.

Other processes affecting the uncertainty budget of ECC ozonesondes such as the different conversion efficiency of sondes from different manufacturers, the uncertainty of the pump efficiency, or a possible temperature dependence of the chemical processes have not been considered here. These effects need to be studied separately; however, they do require the recognition that the time dependence of the chemistry plays an important role in calculating the concentration of ozone under realistic, i.e. non steady state conditions.

**Acknowledgements**

This material is based upon work supported by the National Center for Atmospheric Research, which is a major facility sponsored by the National Science Foundation under Cooperative Agreement No. 1852977. The authors would like to acknowledge funding provided by NASA under contracts NNX17AE37G and NNX17AE41G. The ozonesonde soundings at Costa Rica are currently launched under the skillful

supervision of Dr. Jorge Andres Diaz of the Universidad de Costa Rica. These data are part of the Network for the Detection of Atmospheric Composition Change (NDACC) and are publicly available (see





http://www.ndacc.org). The authors would like to acknowledge very helpful comments by Teresa Campos and Robert Stillwell.

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

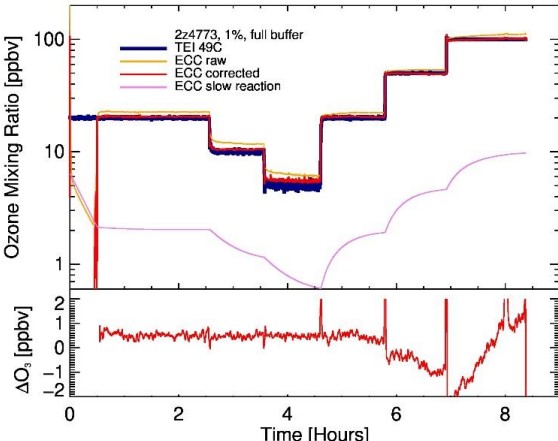

**Figure 1: Top: Ozone mixing ratio generated by the TEI 49C ozone calibrator (blue) and measured by the ECC (original processing in orange, time response corrected in red). The contribution of the slow reaction is shown in purple. Bottom: Difference of corrected ECC measurements and TEI 49C ozone concentration.**


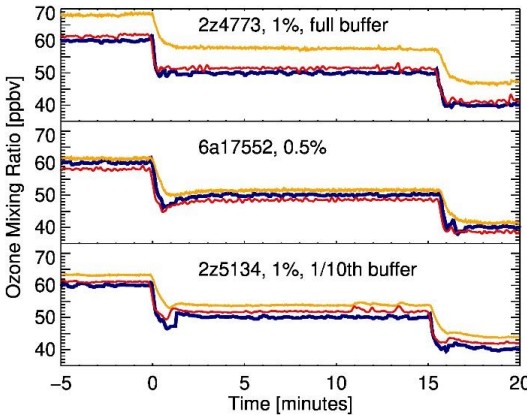

**Figure 2: Response of three ECC sondes using three different solutions during two plateau changes. The color-coding is the same as in Figure 1. The reference time is defined as the time, when the TEI 49C drops below 59 ppb during the first change of plateaus.**


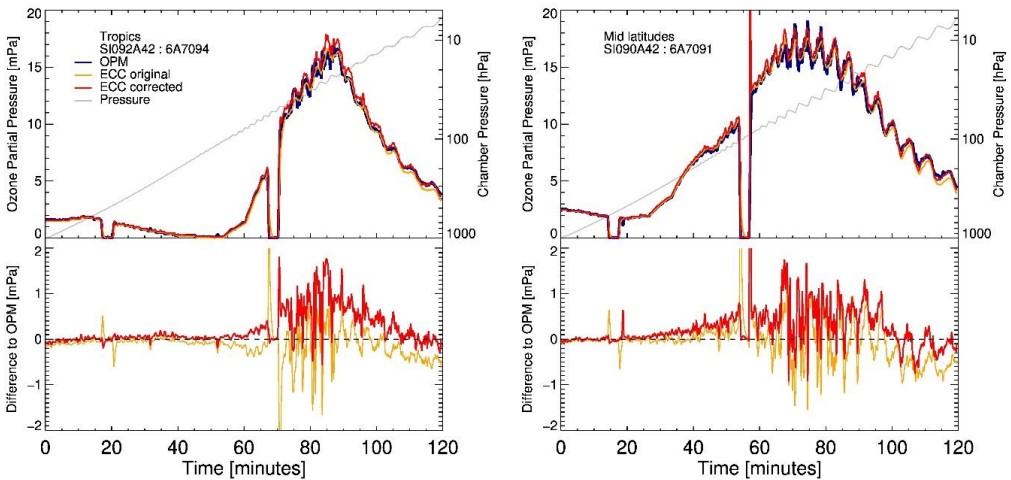

**Figure 3: Reprocessing of JOSIE 2000 environmental simulation chamber ozonesonde data. Left: Tropical simulation; right: mid-latitude simulation. Blue lines in top panels: ozone photometer measurements. Orange lines: originally processed ozonesonde measurements. Red lines: Reprocessed ozonesonde measurements using the separation of slow and fast reaction contribution. Thin grey line: chamber pressure.**

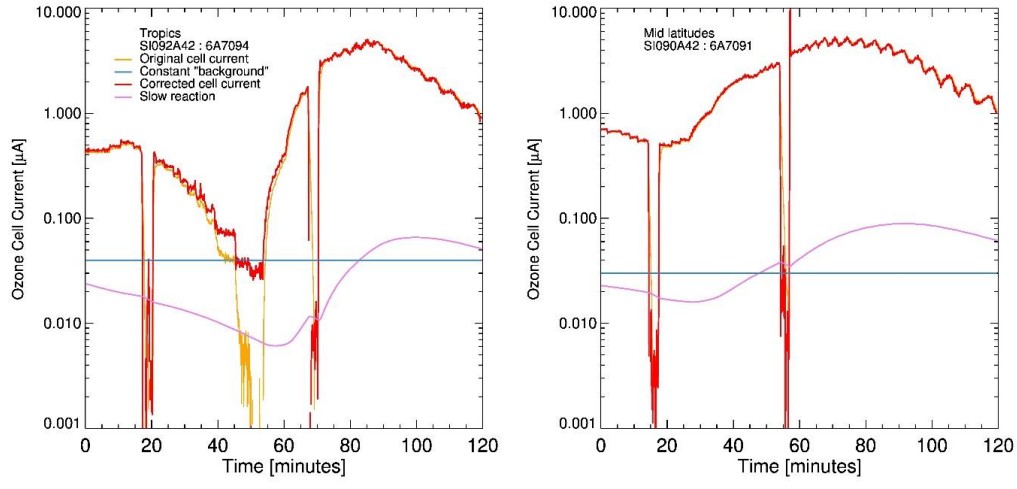

**Figure 4: Cell current components of the tropical and mid-latitude simulations shown in Figure 3. Red lines: Fast cell current contribution. Puprle lines: Slow reaction contribution. Orange lines: Originally measured cell current minus constant background. Light blue lines: Constant background.**






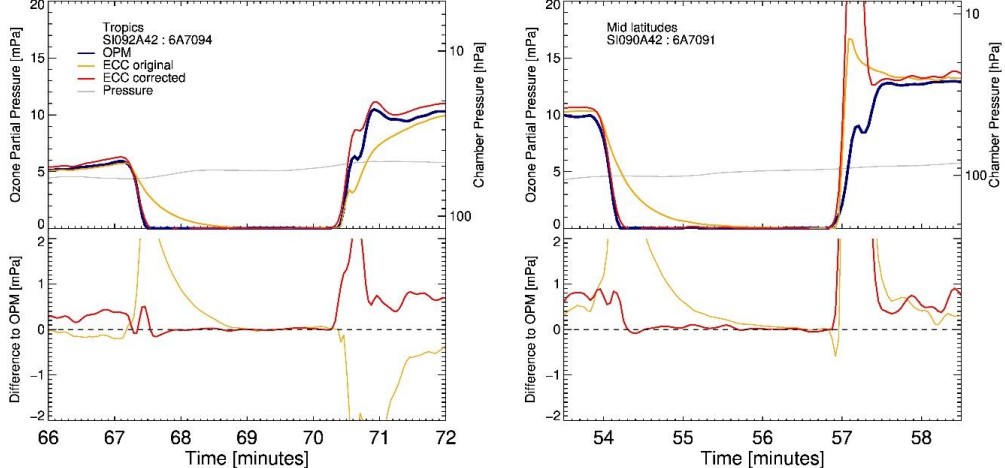

**Figure 5: Same data as Figure 3 showing the time response periods after about 1 hour. Left: Tropical simulation; Right: Mid-latitude simulation. The differences are shown as absolute differences, since the reference achieves zero ozone.**

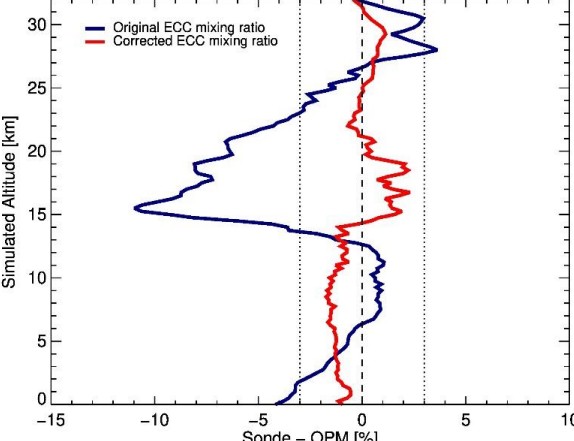


**Figure 6: Comparison between ECC and OPM mixing ratio in 77 simulation experiments during JOSIE 2017. The originally reported difference is shown in blue; the difference calculated using the corrected data is shown in red. Dotted lines indicate ±3%.**






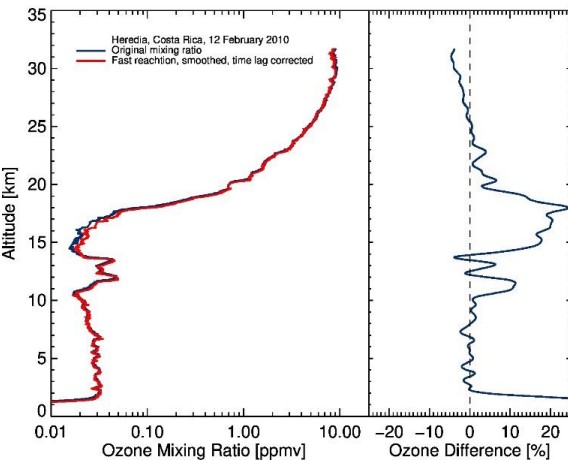

**Figure 7: Ozone profile measured at Costa Rica. The original profile is shown in blue, the reprocessed profile in red. The right hand profile shows the difference of the reprocessed profile minus the original profile.**


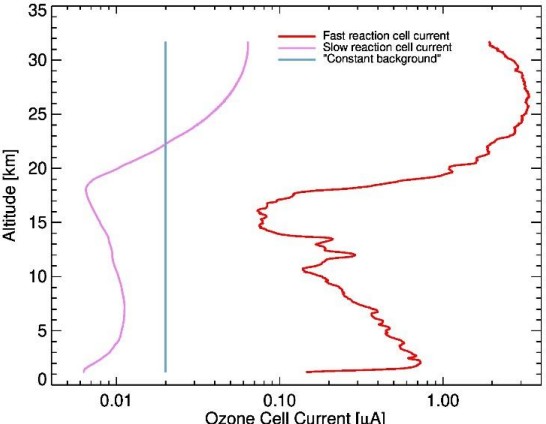

**Figure 8: Contribution of the fast reaction path (red) and the slow reaction path (purple) to the measured cell current. The constant background used in the original processing is shown for reference.**






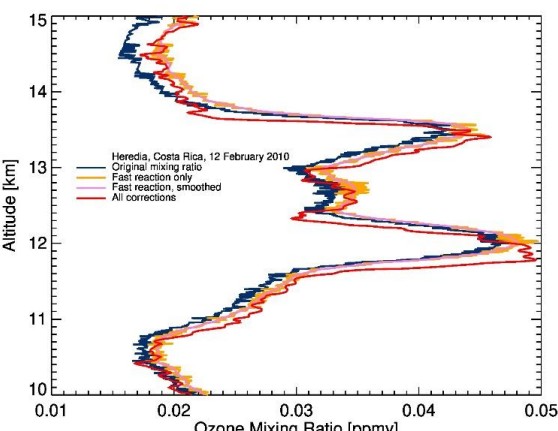

Figure 9: Tropospheric detail of the ozone profile shown in Figure 7.

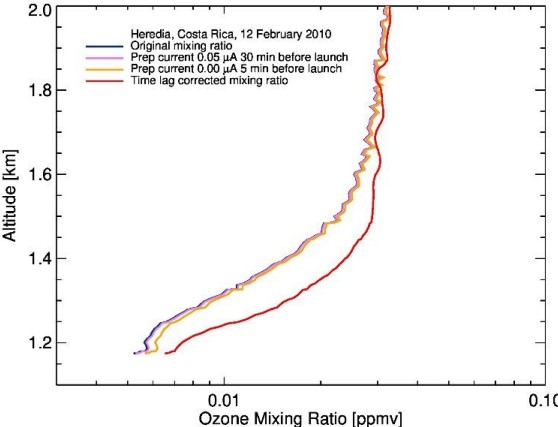

Figure 10: Boundary layer detail of the ozone profile shown in Figure 7. The different assumptions for the preparation
current prior to launch have only been corrected for the slow reaction path, but not time lag corrected for the fast reaction
path.






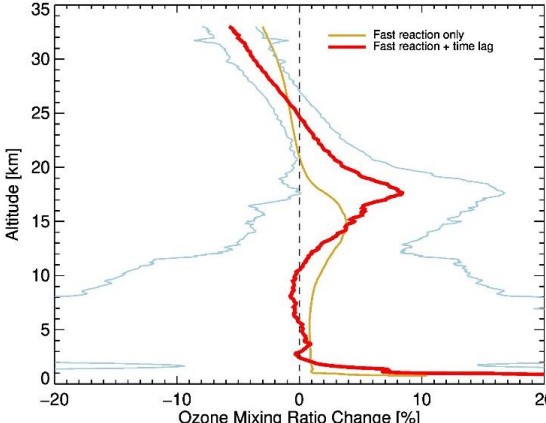

**Figure 11: Median difference between corrected and originally reported ozone mixing ratio for 577 ozonesondes launched at Costa Rica. Thin orange line: removal of the slow reaction contribution only. Thick red line: complete correction algorithm including the time lag correction. The thin blue lines: One standard deviation around the complete correction algorithm.**


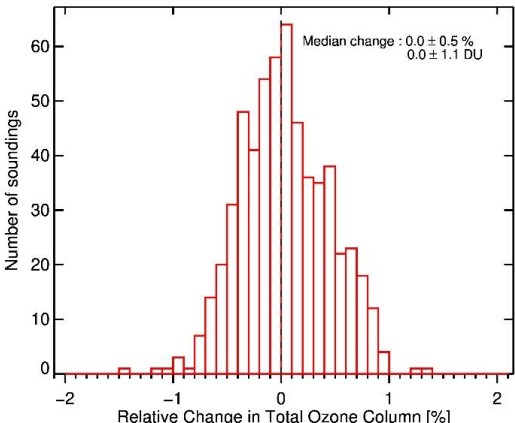

**Figure 12: Change in the total ozone column due to the correction algorithms.**





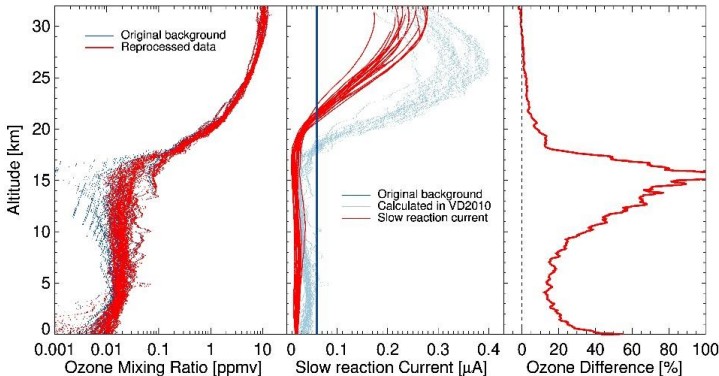

**Figure 13: Left: Original and reprocessed CEPEX ozonesonde profiles. Center: Original constant background, reprocessed background following VD2010 and slow reaction contribution. Right: Difference between reprocessed and original ozonesonde profiles.**

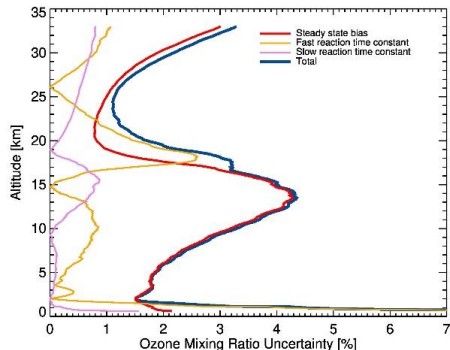

**Figure 14: Contributions to the total uncertainty introduced by the correction shown in Figure 11.**