# Peer review of "A new method to correct the ECC ozone sonde time response and its implications for "background current" and pump efficiency"

_Atmospheric Measurement Techniques, 2020_

## Referee Comment (RC1) · Anonymous Referee #1 · 29 Apr 2020

The issue of the 'background current' which is conventionally subtracted from the measured current in an ozonesonde ascent has been the source of controversy for a long time, with evidence from many sources that it is not, in fact a constant. In this paper the authors argue that the signal measured by an ozonesonde is the sum of a 'fast' component, which is its response to the ambient ozone concentration, and a 'slow' component which arises from other reactions in the cell. A method is described to remove the slow component, and to correct the 'fast' component for lags in the response. Evidence is presented from laboratory and chamber measurements that the new method works very well and improves the accuracy of the ozonesonde measurements. The paper than applies the method to old ozonesonde profiles to evaluate the difference it makes

in practice.

The paper is carefully argued, generally well written, and should be published in AMT with some minor corrections. Before listing these, I would like to invite the authors to consider the following point

The background current problem arises in large part from the practice of exposing the ECC cell to ozone during the preparation. Despite the elegance of the method presented here, there is still some uncertainty in the measurement arising from uncertainty in I'(t0) (eqn 6). Would it not be better to keep this value as small as possible by not exposing the cell to ozone during the preparation?

Points to address:

1. L.170. Equation 4 is not, as stated, a generalisation of equation 3 and the terminology is confusing. I0 and I0' in equation 3 are both constants but in equation 4 they are functions of time. Furthermore the terms Iss and Iss' are introduced without explanation. More care is needed in introducing equation 4.

2. I find the discussion on pp 6-7 of the method to determine the slow reaction term I0'(t) confusing. To get from equation 4 to equation 5 (and hence 6), Iss' is taken as constant. Yet in equation 6, it is replaced by a scaled version of the measured current, which necessarily varies with time. This invalidates the derivation of the equation! Furthermore, the whole point of the slow term is that it is a response to exposure to ozone in the past, so I do not understand how it can be represented by a term proportional only to the measurement at time t. The use of the word 'integration' on line 203 suggests that there may be more to the calculation of the slow pathway than simply plugging numbers into equation 6, but this is how I understand this paragraph. To take an extreme case, an ozonesonde in the tropics encountering a filament of high-ozone air in the lower troposphere then entering a layer of very low ozone concentration just below the tropopause would 'remember' being exposed to ozone in the preparation but not to its much more recent exposure during the profile. In that case equation 6

as written would give too small a slow reaction term IÂň0' and overestimate the ozone concentration.

3. In figure 2, what is the cause of the enormous error in the orange line in the top panel? Was the background current excessively large for this sonde?

4. L.285 concentration (not concentrations)

5. Fig 4 caption, purple

---

## Referee Comment (RC2) · Anonymous Referee #2 · 1 May 2020

**1   Overall remarks**

This is an important paper, providing background on long-standing issues with electro-chemical ozone sondes, offering a viable solution for correcting known problems, and improving the accuracy of ozone profiles measured by these ozone sondes. The paper is generally sound and reasonably well written. It fits well into the scope of AMT. However, because it is a very important paper, I think more effort should be put into clarifying the underlying equations and into presenting the results in a more logical and suggestive flow, and with fewer repetitions and duplications in the text. See my recommendations below. Once these issues are addressed, I think the paper will be a major

contribution to the ozone sounding field. Because it is such an important paper, I hope the authors will put in the additional effort, and will try to streamline their final text.

**2 Major suggestions**

Abstract: It would be more logical to first present the main reaction, which generates 2 electrons per ozone molecule with the fast 20 sec time constant (and has been known forever). Then present the slow reaction with 25 min time constant, that produces between 0.05 to 0.2 electrons per ozone molecule (2 electrons $\times \alpha$, from the numbers given near Equation 6). The major innovation of the paper is the characterization of this slow reaction, but overall it is still a secondary reaction. So I suggest to switch the order in which the reactions are presented, and also mention how many electrons the slow reaction generates (compared to 2 electrons from the fast main reaction).

Summary: The summary is very well written!! Could the person writing the summary please try to remove duplications and repetitions in the main text, and make the main text more logically flowing and succinct?

Equation 1 and following discussion: I am not happy about Equation 1. Pump correction, ozone to electrons conversion efficiency, hysteresis and background current effects are all lumped together in this empirical $\gamma$ correction (or fudge factor). Later in the text (lines 114 to 138) the authors bend over backwards, explaining that use of the correct Johnson et al. 2002 pump corrections (not the fudged Khomhyr et al. ones) and proper accounting for secondary slow reactions are the correct way to go. To me, it would make a lot of sense to separate the different "$\gamma$s" here, and to properly introduce and explain them, $\gamma_{pump}$, $\gamma_{conversion}$, .... The text from lines 114 to 138 should be moved here as well. In the end, according to the paper, the fudged $\gamma$s can be unfudged and only the correct $\gamma_{pump}$ is required. Accounting for the slow secondary reaction takes care of the rest / the background.

After properly introducing and explaining the different "$\gamma$s" near Equation 1 already, the rest of the discussion would then focus on background alone, with less jumping forth and back between background, and conversion efficiency / solutions. After moving the text parts on solutions and $\gamma$s, as suggested above and also further below, the authors should then revisit their discussion of background currents, background subtraction practices, and background experiments (lines 70 to 123). Make it more concise and more logically flowing. Right now, it is a lot of forth and back. E.g. the paragraph around line 110 seems out of place where it is, and should probably come earlier.

**3   Mathematics**

Just like reviewer 1, I cannot follow the mathematical reasoning from Equation 3 to 6. I find Equation 4 plausible, although it is not really clear what $I_{ss}$ and $I'_{ss}$ are. They seem to drop from the sky. I assume they are something like the real $I_{O3}$ and $I'_{O3}$ that would be measured, if the time constants of the fast and slow reactions were infinitely small. I do not understand how the authors get from Equation 4 to Equation 5. Taking the derivative of Equation 5, I get something very different from Equation 4 (even when I only take the slow reaction part of Equation 4 ($\frac{dI'_0}{dt} = -\frac{1}{\tau'}(I'_0 - I'_{ss})$ ).

What I could understand, is using the slow part of Equation 4 to numerically calculate $I'_0$ from one time step to the next:

$$I'_0(t) = I'_0(t_0) - \frac{t - t_0}{\tau'} \left( I'_0(t_0) - I'_{ss}(t_0) \right) \tag{1}$$

This can be re-arranged to

$$I'_0(t) = \frac{t - t_0}{\tau'} I'_{ss}(t_0) + \left( 1 - \frac{t - t_0}{\tau'} \right) I'_0(t_0) \tag{2}$$

which is the first order term of a Taylor series expansion of the authors' Equation 5, and is probably the numerical solution the authors are using anyway. Assuming $I'_{ss} = \alpha I_{ss} = \alpha I_{O3} \approx \alpha I_m$ we would then arrive at the first order term of the Taylor series expansion of the authors' Equation 6.

$$I'_0(t) = \frac{t - t_0}{\tau'}\alpha I_m(t_0) + \left(1 - \frac{t - t_0}{\tau'}\right) I'_0(t_0) \tag{3}$$

To me this seems a more logical and mathematically stringent way. For small time steps (seconds, compared to the 25 minute time constant), this Equation 3 will give very similar results to the authors Equation 6. However, both reviewer 1 and I were not able to understand how the authors' Equations 5 and 6 were derived, and we both are wondering if they are correct. Given all this, I suggest that the authors check their mathematical reasoning, consider using the simpler linear equations above, and re-think and simplify the accompanying text.

By the way: Reviewer 1 was wondering why in Equation 6 $I'_0(t)$ only depends on the instantaneous $I_m(t)$, and does not integrate $I_m(t)$ over time. The iterative process described by Equation 3 above clearly shows that $I'_0(t)$ integrates over $I_m$ from all previous time-steps.

Since the authors' Equation 3 only seems to inspire Equation 4, but is not really mathematically connected here, I sugggest to drop it completely. It seems more confusing then helpful here. Just introduce the two time constants, and start with Equation 4.

**4 Minor suggestions**

Line 22: insert "secondary" before "slow reaction". Switch order of the two reactions, as mentioned above.

Line 25: "remaining" -> "main".

Line 37: A pump correction cannot improve the time response. In fact no post-processing can change the time response, which is defined by hardware and chemistry. Instead of "which . . . ", start a new sentence. "This improves the accuracy of ECC sonde ozone profiles, especially for very low ozone cooncentrations or large ozone gradients, and removes systematic biases . . . "

Lines 39 to 41: I don't get this entire sentence. 1.) ECC measurements are influenced, no matter how you correct for it. So this is really a NULL statement. 2.) You already said in the previous paragraph that, due to correcting for the fast response time, gradients are reported more accurately. So what is NEW here? I would suggest dropping the entire sentence, or really making clearer what is NEW and IMPORTANT here.

Line 59: As mentioned above, most operationally used $\gamma$s are more than a pump correction. So I think this text should be corrected here, and clarified.

Lines 68-72: It would be good to give some numbers / ranges for typical background currents here.

Lines 91 to 102: I think the introduction of the different solutions here interrupts the logical flow of the background discussion. Therefore, I suggest to move this paragraph right after line 50, before Equation 1.

Lines 105/106: Fix "assumption" . . . "assumed"

Line 107: "zero" missing before "ozone"?

Lines 123 to 138: As already mentioned, I think this should be disussed earlier, right after Equation 1 when $\gamma$ is discussed.

Line 133: "currently recognized as properly describing" -> "properly describes . . . and is consistent . . . "?

Equations 10 and 11: Minus sign is missing in the exponents.

Figure 1: Maybe also show the difference between ECC raw and TEI49C in the bottom panel. That would show the improvement by the new processing even better.

Figure 2: It would be good to also show the derived slow reaction current, on a log-scale, or in a separate panel.

Line 256: I do not understand how 6 step changes can represent a total of 60 step changes. Do you mean average of 60 step changes? Or just 6 (typical) examples? Please reword / clarify.

Figure 3 / Line 291: "follow slightly better". Line 319: "dramatically" improved. I don't know. To me it rather looks like the yellow (raw ECC) curves in the bottom panels are overall closer to the zero lines than the red (ECC corrected) curves. I certainly do not see a "dramatic" improvement. The wording should be more neutral / conservative here.

Figure 4: Can you please also show the correction for the fast (25 sec) response time? It seems to be much smaller than the pink correction for the slow (25 min) reaction.

Lines 294 to 303, Figs. 3 and 4: So the pump corrections were also changed. Now things get really confusing. Can you please dis-entangle the pump correction effect? Show with an additional separate line? Do not change the pump correction here?

Figure 5: OK, here I can see an improvement. Most of it seems to come from the "fast response" correction.

Figure 6: Several additional things would be good here. a.) standard deviations for the red and blue difference profiles. b.) the average ozone profiles (original, corrected and OPM reference. c.) the average fast and slow response corrections.

Lines 354 to 356: This has already been said many times. Delete the sentence.

Figure 7: I assume there was no change in the pump correction - different from Figs. 3 and 4.

Line 362: Delete "what was assumed with"?

Line 366: Replace "smoothing and correcting for time lag" by "fast response correction"?

Line 374: Replace "The" by "This"?

Line 392: "rapidly decays"? Should it not decay with a time constant of 25 minutes? Maybe reword "is about 10% at launch, but soon becomes insignificant, as ozone is increasing rapidly with altitude."

Figure 11: Should "fast reaction" in the legend in the plot not be "slow reaction"? Figure caption and legend in the plot seem to be inconsistent.

Line 437: Add "much" after "affected"? There is an effect, up to about 1% in Fig. 12.

Line 439, Figure 13: How many ozone sondes during CEPEX? Maybe add the number. Rightmost panel: Maybe add standard deviations, or thin lines of individual profiles? Why is the peak change so much larger ($\approx$80%) than in Fig. 11 (8%)?

Lines 467 to 477: Has that not been said several times by now? Maybe drop these two paragraphs?

Line 483, see also line 392: Well, the effect should be there for 10 to 20 minutes. However, if ozone becomes large enough it will be very small. Reword?

Line 517: "justify" seems the wrong word. "quantify"?

Lines 516 to 524: There is a lot hedging, ifs, and maybes here. Can this not be worded shorter and more succinct? E.g. The two sentences in lines 521 to 524 are saying alsmost the same thing.

Lines 544 to 548: This has been known and sometimes corrected since the 1980s. Maybe add one or two of these old references?

---

## Short Comment (SC1) · 5 May 2020

I have read through this paper manuscript as well as the two referee reports with great interest. I think that this is indeed a very interesting and important paper. I have some comments, which are written below.

(1) The "time lag correction" for the fast reaction pathway

Although the Introduction has extensive discussion on the "background current" issue and the proposed, appropriate treatment for the slow reaction pathway, large changes/improvements are actually seen by applying the "time lag correction" for the

fast reaction pathway, i.e., in Figures 5, 6, 9, and 11. In particular, we see large changes with opposite signs in the lower stratosphere and in the upper stratosphere due to the opposite signs in the average vertical gradient of ozone. In Figure 9, the effects of the time lag correction are seen also at the edges of tropical tropospheric ozone enhanced layers. In Figure 10, at the first ~400 meters from the surface, it seems to me that the large changes are again due to this time lag correction.

Therefore, I think that a review on this time lag correction needs to be made also in the Introduction. I strongly believe that the Section 4 of Imai et al. (JGR, 2013) (I am a coauthor of this paper) is one of the appropriate and very useful references for this purpose. This Imai et al. paper explains the mathematics of the time lag correction procedure, and shows the effects of this correction on worldwide ozonesonde sounding data in the stratosphere. Their example of ascent vs. (much faster) descent comparison in a tropical sounding would also be useful to understand the effects. The results in Figure 10 (bottom left for 15N-15S) of their paper agree quite well with what is shown in Figure 11 (as well as Figure 6) of this Voemel et al. manuscript (note the difference in the definition of the difference (the variable for the x axis)).

To my knowledge, the paper by Imai et al. was the first to explicitly show the effects of this response time issue in ozonesonde data, but Referee #2 noted, "This (i.e., the time lag of the fast reaction pathway) has been known and sometimes corrected since the 1980s. Maybe add one or two of these old references?" . . . I am also interested in those old papers.

We (Masato Shiotani, Naohiro Manago, and myself) wanted to write a follow-on paper on this ozonesonde response time issue using JOSIE data, but we found that the temporal resolution of currently available JOSIE data is not enough high for quantitative analysis. Some of our preliminary results were presented in the SPARC General Assembly (Kyoto, Japan) in 2018 and in other meetings.

(2) The method and the terminology

[Figure]

I totally agree with the two referees that the mathematics need major revisions. All the symbols should be clearly defined. In addition, I also would like to emphasize that all the legends in the figures should be carefully revised, so that the same terminology, as will be re-defined in the Method section, is used; the legends in different figures are currently not consistent and thus are a source of confusion.

Below, let me write my understanding of the Method after reading through the Application . . . and Discussion sections.

For the conventional data processing, the cell current corresponding to the ambient ozone concentration, $I_a(t)$, is calculated as:

$$I_a(t) = I_m(t) - I_{bc}$$

where $I_m(t)$ is measured cell current, and $I_{bc}$ is the "background current" which is usually treated as a constant. (However, in practice, the conventional "pump efficiency coefficient" (as a function of ambient pressure) has a component which depends on the buffered KI solution recipe, thus, in a sense, providing part of information on the slow response pathway, though it might have been incorrect anyway.)

For the proposed data processing in this paper, the measured cell current $I_m(t)$ is assumed to be a summation of two components as:

$$I_m(t) = I_f(t) + I_s(t)$$

where $I_f(t)$ is due to the fast reaction pathway, and $I_s(t)$ is due to the slow reaction pathway. $I_s(t)$ is, roughly speaking (as I still do not fully understand), assumed to have two components, the "steady state bias" depending on the buffer (and evaluated by VD2010) and the delayed and integrated response with a time scale of $\sim$25 minutes.

First, $I_s(t)$ is evaluated based on the results by VD2010 and by other considerations. In this process, the "post-preparation current" might or might not be used. Then, we obtain $I_f(t) = I_m(t) - I_s(t)$.

I_a (t) is then obtained from I_f (t) by applying the "time lag correction" with the time scale of ∼20 seconds. (I would like to note that the smoothing with the Gaussian filter used by the authors is not conceptually essential, but practically necessary, because noises in the original data can be amplified when the time lag correction is applied. Thus, separating this from the discussion of the concept may be good for the readers. Also, the "Fast reaction, smoothed" line in Figure 9 might not be essential for that figure.)

Please correct me if my understanding of the Method is not correct.

Again, in the paper, including the figure legends, please use exactly the same words (or even the defined symbol) for the same variable.

Written by Masatomo Fujiwara

References:

Imai, K., M. Fujiwara, Y. Inai, N. Manago, M. Suzuki, T. Sano, C. Mitsuda, Y. Naito, F. Hasebe, T. Koide, and M. Shiotani (2013), Comparison of ozone profiles between Superconducting Submillimeter-Wave Limb-Emission Sounder and worldwide ozonesonde measurements, Journal of Geophysical Research, 118(22), 12755-12765, doi: 10.1002/2013JD021094.

---

## Author Comment (AC1) · 30 Jul 2020

**Replies to reviewers**

We appreciate the very helpful comments from the reviewers and our colleague Prof. Fujiwara. We have revised the manuscript following their suggestions. The detailed replies to specific comments are shown below indented and in italics. Highlighted text changes are included in the replies in quotes.

**Reviewer 1**

The background current problem arises in large part from the practice of exposing the ECC cell to ozone during the preparation. Despite the elegance of the method presented here, there is still some uncertainty in the measurement arising from uncertainty in I'(t0) (eqn 6). Would it not be better to keep this value as small as possible by not exposing the cell to ozone during the preparation?

> *While the reviewer is correct that less exposure to ozone prior to launch would reduce the build up of the secondary slow reaction, exposure to ozone is an essential step in the standard operating procedures to test the functionality of the sonde and to measure the fast reaction time constant. To address the uncertainty of the slow reaction pathway prior to launch, the revised standard operating procedures will pay more attention to the time between ozone exposure during the preparation and launch to allow the slow reaction to decay sufficiently.*

1. L.170. Equation 4 is not, as stated, a generalisation of equation 3 and the terminology is confusing. I0 and I0' in equation 3 are both constants but in equation 4 they are functions of time. Furthermore the terms Iss and Iss' are introduced without explanation. More care is needed in introducing equation 4.

2. I find the discussion on pp 6-7 of the method to determine the slow reaction term I0'(t) confusing. To get from equation 4 to equation 5 (and hence 6), Iss' is taken as constant. Yet in equation 6, it is replaced by a scaled version of the measured current, which necessarily varies with time. This invalidates the derivation of the equation! Furthermore, the whole point of the slow term is that it is a response to exposure to ozone in the past, so I do not understand how it can be represented by a term proportional only to the measurement at time t. The use of the word 'integration' on line 203 suggests that there may be more to the calculation of the slow pathway than simply plugging numbers into equation 6, but this is how I understand this paragraph. To take an extreme case, an ozonesonde in the tropics encountering a filament of high ozone air in the lower troposphere then entering a layer of very low ozone concentration just below the tropopause would 'remember' being exposed to ozone in the preparation but not to its much more recent exposure during the profile. In that case equation 6 as written would give too small a slow reaction term IÂn0' and overestimate the ozoneˇ concentration.

> *We have taken both reviewers confusion about the derivation of the time response corrections expressed here in points 1 and 2 to heart, simplified the explanations, and slightly changed the terminology to make it easier to follow. We removed Equation 3 completely and started with a simplified version of Equation 4 to express that the measured cell current is a superposition of the slow and fast reaction pathway. That the time constant of the slow reaction is about 75 times larger than that of the fast reaction allows treating the slow pathway separately. Calculating the slow reaction pathway is effectively done iteratively over short time periods during which the assumption of equilibrium state may be considered reasonably close. The reviewer is correct and in a strict sense, replacing the equilibrium state of the slow reaction path with a small fraction of*

*the measured cell current is not correct. Even if hypothetically a rising ozone sonde would come to a complete stop and measure a constant ozone concentration, both the time response of the slow and fast reaction will slightly change the measured cell current. However, that change is small compared to the overall magnitude of the measured cell current and the associated error of the correction small. This could be improved by iterating the process; however, it is questionable, whether this would lead to a further improvement.*

*Our explanations did not make it clear that the iterative algorithm is effectively an integration over the observed ozone concentration over the past slow time constants. We hope that the revised explanations make this point clearer.*

3. In figure 2, what is the cause of the enormous error in the orange line in the top panel? Was the background current excessively large for this sonde?

*Many previous studies in the past, most importantly the paper by Johnson et al. (2002) have shown that the 1% full buffer solution substantially overestimates the amount of ozone. The top panel in Figure 2 is another example and emphasizes this fact, since it is processed without any pump efficiency correction (measurements were done in the lab) and without "background".*

4. L.285 concentration (not concentrations)

*Corrected*

5. Fig 4 caption, purple

*Corrected*

**Reviewer 2**

**Major suggestions**

Abstract: It would be more logical to first present the main reaction, which generates 2 electrons per ozone molecule with the fast 20 sec time constant (and has been known forever). Then present the slow reaction with 25 min time constant, that produces between 0.05 to 0.2 electrons per ozone molecule (2 electrons ×α, from the numbers given near Equation 6). The major innovation of the paper is the characterization of this slow reaction, but overall it is still a secondary reaction. So I suggest to switch the order in which the reactions are presented, and also mention how many electrons the slow reaction generates (compared to 2 electrons from the fast main reaction).

> *We have changed the abstract and switched these two sentences. They now read:*

> *'The main fast reaction pathway with a time constant of about 20 s is due the conversion of iodide to molecular iodine and the generation of two free electrons per ozone molecule. A slow reaction pathway involving the buffer generates an excess current of about 2% – 10% with a time constant of about 25 min can be interpreted as what has conventionally been considered the "background current".'*

Summary: The summary is very well written!! Could the person writing the summary please try to remove duplications and repetitions in the main text, and make the main text more logically flowing and succinct?

> *We have worked through the text, removed repetitions, and made it more precise. See below for details.*

Equation 1 and following discussion: I am not happy about Equation 1. Pump correction, ozone to electrons conversion efficiency, hysteresis and background current effects are all lumped together in this empirical γ correction (or fudge factor). Later in the text (lines 114 to 138) the authors bend over backwards, explaining that use of the correct Johnson et al. 2002 pump corrections (not the fudged Khomhyr et al. ones) and proper accounting for secondary slow reactions are the correct way to go. To me, it would make a lot of sense to separate the different "$\gamma_s$" here, and to properly introduce and explain them, $\gamma_{pump}$, $\gamma_{conversion}$, . . . . The text from lines 114 to 138 should be moved here as well. In the end, according to the paper, the fudged $\gamma_s$ can be unfudged and only the correct $\gamma_{pump}$ is required. Accounting for the slow secondary reaction takes care of the rest / the background.

> *Although the reviewer is correct that other efficiencies play a role, these are considered small. The GAW report #201 explicitly only considers an additional conversion efficiency but states that $\gamma_{conversion} \approx 1$. Efficiency corrections such as that for the uptake of ozone in smaller amounts of cathode solution or the efficiency of the different manufacturers are either limited in scope (solution amount difference is relevant mostly for the Canadian network) or not as well defined (manufacturer difference). On the other hand, the pump efficiency correction factors in use range between 5 and 14 % at 10 hPa. This ambiguity is dominating the error budget at that pressure. Most importantly, the current recommendation of matching pump efficiency correction with a particular solution is historically explained, but scientifically not appropriate. Therefore, we decided to ignore the minor corrections, which are on the order of one and focus only on the dominating "fudged" efficiency correction, i.e. the pump efficiency. We decided not to expand the discussion of the minor efficiency corrections to keep the manuscript focused.*

After properly introducing and explaining the different "$\gamma_s$" near Equation 1 already, the rest of the discussion would then focus on background alone, with less jumping forth and back between background, and conversion efficiency / solutions. After moving the text parts on solutions and $\gamma_s$, as suggested above and also further below, the authors should then revisit their discussion of background currents, background subtraction practices, and background experiments (lines 70 to 123). Make it more concise and more logically flowing. Right now, it is a lot of forth and back. E.g. the paragraph around line 110 seems out of place where it is, and should probably come earlier.

> *We have followed the reviewer's suggestion and reordered the text and improved the logical flow. See the details below.*

**Mathematics**

Just like reviewer 1, I cannot follow the mathematical reasoning from Equation 3 to 6.

I find Equation 4 plausible, although it is not really clear what $I_{ss}$ and $I'_{ss}$ are. They seem to drop from the sky. I assume they are something like the real $I_{O3}$ and $I'_{O3}$ that would be measured, if the time constants of the fast and slow reactions were infinitely small. I do not understand how the authors get from Equation 4 to Equation 5. Taking the derivative of Equation 5, I get something very different from Equation 4 (even when I only take the slow reaction part of Equation 4 $\frac{dI'_0}{dt} = -\frac{1}{\tau'}(I'_0 - I'_{ss})$

What I could understand, is using the slow part of Equation 4 to numerically calculate $I'_0$ from one time step to the next:

$$I'_0(t) = I'_0(t_0) - \frac{t-t_0}{\tau'}\left(I'_0(t_0) - I'_{ss}(t_0)\right)$$

(1)

This can be re-arranged to

$$I'_0(t) = \frac{t-t_0}{\tau'}I'_{ss}(t_0) + \left(1 - \frac{t-t_0}{\tau'}\right)I'_0(t_0)$$

which is the first order term of a Taylor series expansion of the authors' Equation 5, and is probably the numerical solution the authors are using anyway. Assuming $I'_{ss} = \alpha I_{ss} \approx \alpha I_m$ we would then arrive at the first order term of the Taylor series expansion of the authors' Equation 6.

$$I'_0(t) = \frac{t-t_0}{\tau'}\alpha I_m(t_0) + (1 - \frac{t-t_0}{\tau'})I'_0(t_0)$$

(3)

To me this seems a more logical and mathematically stringent way. For small time steps (seconds, compared to the 25 minute time constant), this Equation 3 will give very similar results to the authors Equation 6. However, both reviewer 1 and I were not able to understand how the authors' Equations 5 and 6 were derived, and we both are wondering if they are correct. Given all this, I suggest that the authors check their mathematical reasoning, consider using the simpler linear equations above, and rethink and simplify the accompanying text.

By the way: Reviewer 1 was wondering why in Equation 6 $I'_0(t_0)$ only depends on the instantaneous $I_m(t)$, and does not integrate $I_m(t)$ over time. The iterative process described by Equation 3 above clearly shows that $I'_0(t)$ integrates over $I_m$ from all previous time-steps.

Since the authors' Equation 3 only seems to inspire Equation 4, but is not really mathematically connected here, I sugggest to drop it completely. It seems more confusing then helpful here. Just introduce the two time constants, and start with Equation 4.

> *We followed the reviewer's advice, removed Equation 3, and started with a simplified Equation 4. Please see our reply to the similar points raised by reviewer 1, which led us to rework the explanations of the algorithm.*

**Minor suggestions**

Line 22: insert "secondary" before "slow reaction". Switch order of the two reactions, as mentioned above.

> *added*

Line 25: "remaining" -> "main".
> *changed*

Line 37: A pump correction cannot improve the time response. In fact no postprocessing can change the time response, which is defined by hardware and chemistry. Instead of "which . . . ", start a new sentence. "This improves the accuracy of ECC sonde ozone profiles, especially for very low ozone cooncentrations or large ozone gradients, and removes systematic biases . . . "

> *changed*

Lines 39 to 41: I don't get this entire sentence. 1.) ECC measurements are influenced, no matter how you correct for it. So this is really a NULL statement. 2.) You already said in the previous paragraph that, due to correcting for the fast response time, gradients are reported more accurately. So what is NEW here? I would suggest dropping the entire sentence, or really making clearer what is NEW and IMPORTANT here.

> *We have rephrased this sentence as*

> *"In the surface layer, operational procedures prior to launch, in particular the use of filters, influence how typical gradients above the surface are detected. The correction algorithm may report gradients that are steeper than originally reported, but their uncertainty is strongly influenced by the pre-launch procedures."*

> *The important point here is that the data typically do not contain, when a filter was removed from a sonde prior to launch and what time has elapsed between sonde conditioning and launch. In the algorithm, a value for the slow reaction pathway has to be prescribed, the contribution of which decays slowly as the sonde rises. Above the middle troposphere, these factors no longer play a significant role.*

Line 59: As mentioned above, most operationally used $\gamma_s$ are more than a pump correction. So I think this text should be corrected here, and clarified.

> *As mentioned above, we decided to focus on the dominant efficiency correction. A discussion of the minor efficiency corrections would be distracting.*

Lines 68-72: It would be good to give some numbers / ranges for typical background currents here.

> *We added a typical range of about 0.01 to 0.05 µA.*

Lines 91 to 102: I think the introduction of the different solutions here interrupts the logical flow of the background discussion. Therefore, I suggest to move this paragraph right after line 50, before Equation 1.

*We moved this paragraph right after Equation (1).*

Lines 105/106: Fix "assumption" . . . "assumed" Line 107: "zero" missing before "ozone"?

*Changed the sentence to "It cannot be assumed that these filters operate with perfect efficiency and under all conditions."*

Line 107: "zero" missing before "ozone"?

*Here we refer to the exposure of ozone generating 5 μA and more specifically the following minutes, when typically a filter is used to generate ozone free air.*

Lines 123 to 138: As already mentioned, I think this should be disussed earlier, right after Equation 1 when γ is discussed.

*We moved this paragraph up.*

Line 133: "currently recognized as properly describing" -> "properly describes . . . and is consistent . . . "?

*Changed this sentence to "However, only the pump flow correction by Johnson et al. (2002) currently describes the loss of pump efficiency consistent with measurements from other groups (Tatsumi Nakano, personal communication)."*

Equations 10 and 11: Minus sign is missing in the exponents.

*The minus sign was present but the type setting merged it with the fraction. This is now fixed.*

Figure 1: Maybe also show the difference between ECC raw and TEI49C in the bottom panel. That would show the improvement by the new processing even better.

*We included the difference between the raw ECC and the TEI49C in the bottom panel. We had to change the vertical axis scaling to fit this curve onto the same plot, which makes the point even stronger.*

Figure 2: It would be good to also show the derived slow reaction current, on a logscale, or in a separate panel.

*The main point of Figure 2 is to show the similar behavior of the three different solutions. We decided not to include the slow reaction current in this Figure because it would make the Figure too large without adding information that is not already contained in Figure 1.*

Line 256: I do not understand how 6 step changes can represent a total of 60 step changes. Do you mean average of 60 step changes? Or just 6 (typical) examples? Please reword / clarify.

*Changed sentence to "The time lag corrections for the six step changes shown in Figure 2 are a representative subset of a total of 60 step changes in 25 different experiments."*

Figure 3 / Line 291: "follow slightly better". Line 319: "dramatically" improved. I don't know. To me it rather looks like the yellow (raw ECC) curves in the bottom panels are overall closer to the zero lines than the red (ECC corrected) curves. I certainly do not see a "dramatic" improvement. The wording should be more neutral / conservative here.

*We removed the word "dramatic". Although the mean comparison of the raw and the reference may appear better for the tropical simulation, the pressure controller induced ozone oscillations inside the chamber are clearly better represented in the corrected profiles of both simulations.*

Figure 4: Can you please also show the correction for the fast (25 sec) response time? It seems to be much smaller than the pink correction for the slow (25 min) reaction.

*We added the fast reaction contribution prior to time lag correction. Not surprising, on this log scale, it is barely distinguishable from the time lag corrected current outside the sharp drop-offs. The main point of this figure is to show that the calculated contribution of the slow reaction varies strongly with time in contrast to a constant background current.*

Lines 294 to 303, Figs. 3 and 4: So the pump corrections were also changed. Now things get really confusing. Can you please dis-entangle the pump correction effect? Show with an additional separate line? Do not change the pump correction here?

*We added a line showing the effects of the different pump efficiency corrections. In this example, the incorrect pump efficiencies produce better results in the ozone peak, but a larger bias at the lowest pressures. The Johnson et al. (2002) pump efficiencies produce a better agreement at the lowest pressures, where the correction is strongest. The effect of the different pump efficiency corrections is explained in the main text.*

Figure 5: OK, here I can see an improvement. Most of it seems to come from the "fast response" correction.

*The zero ozone periods in the JOSIE runs are useful to highlight the effect of the fast response, which becomes most obvious in these phases.*

Figure 6: Several additional things would be good here. a.) standard deviations for the red and blue difference profiles. b.) the average ozone profiles (original, corrected and OPM reference. c.) the average fast and slow response corrections.

*We added the standard error to the average difference profiles. The standard deviation is large in particular in the simulated UTLS region, since the profiles do not line up at the simulated tropopause. We did not show the averaged individual profiles or their components. Examples of the fast and slow components are shown in the previous Figures and don't add much value here. We have update the text to reflect the addition of the standard error:*

*"There are many details in this data set, which are smoothed out by the averaging and require a more detailed analysis, especially in the 12-20 km region, where the standard error is large. Nevertheless, we show that the structure in the difference profile is strongly reduced and that on average the ECC sondes agree with the OPM to well within 3% throughout all pressures."*

Lines 354 to 356: This has already been said many times. Delete the sentence.

*Deleted.*

Figure 7: I assume there was no change in the pump correction - different from Figs. 3 and 4.

*Correct. Costa Rica has used the 1%, 0.1 buffer solutions from the beginning and accordingly the pump efficiency by Johnson et al., (2002).*

Line 362: Delete "what was assumed with"?

*Phrase deleted*

Line 366: Replace "smoothing and correcting for time lag" by "fast response correction"?

*We left this expression in place because we wanted to highlight that this Figure shows the effect of each individual processing step.*

Line 374: Replace "The" by "This"?

*Changed*

Line 392: "rapidly decays"? Should it not decay with a time constant of 25 minutes? Maybe reword "is about 10% at launch, but soon becomes insignificant, as ozone is increasing rapidly with altitude."

*Deleted the word "rapidly"*

Figure 11: Should "fast reaction" in the legend in the plot not be "slow reaction"? Figure caption and legend in the plot seem to be inconsistent.

*Changed "removal of the slow reaction contribution" to "fast reaction contribution" to make caption and legend consistent.*

Line 437: Add "much" after "affected"? There is an effect, up to about 1% in Fig. 12.

*Added "much"*

Line 439, Figure 13: How many ozone sondes during CEPEX? Maybe add the number. Rightmost panel: Maybe add standard deviations, or thin lines of individual profiles? Why is the peak change so much larger (≈80%) than in Fig. 11 (8%)?

*We have included the number of profiles (28) in the text and added the differences of all individual profiles in the right hand panel. The change is so much larger, since the western Pacific region has the lowest ozone concentration in the upper troposphere and lower stratosphere and because the original data were processed with a relatively high background (0.065 μA), which prompted revisiting these data in the first place. Modern standard operating procedures typically lead to much lower "background" currents, making differences such as in Costa Rica less dramatic.*

Lines 467 to 477: Has that not been said several times by now? Maybe drop these two paragraphs?

*At Costa Rica, the Johnson et al. (2002) pump efficiency correction has been used in the original processing and remained unchanged in the reprocessing. In contrast, the CEPEX data used the Komhyr (1986) originally and Johnson et al. (2002) in the reprocessing. Unlike the Costa Rica data, it is the CEPEX data set in which we can show that the slow reaction component provides the proper correction for the excess cell current in the upper part of the profile, allowing to move from the fudged Komhyr corrections to an appropriate pump efficiency correction without the need for another empirical correction. This result is similar to what we found in the JOSIE data and confirmed here in atmospheric measurements. We have decided to keep these paragraphs.*

Line 483, see also line 392: Well, the effect should be there for 10 to 20 minutes. However, if ozone becomes large enough it will be very small. Reword?

> *The surface layer, i.e. the lowest few hundred meter, is most strongly affected and the entire boundary layer is a reasonable estimate for the region, in which the choice of partitioning is significant, even though mathematically, the effect extends well into the free troposphere. The typically larger ozone concentration above the surface reduces the influence of that choice. We have changed the word "only" to "mostly" in the new clause:*

> *"however, this assumption mostly influences the calculated ozone mixing ratio in the boundary layer."*

Line 517: "justify" seems the wrong word. "quantify"?

> *Replaced "may justify" with "help quantifying"*

Lines 516 to 524: There is a lot hedging, ifs, and maybes here. Can this not be worded shorter and more succinct? E.g. The two sentences in lines 521 to 524 are saying almost the same thing.

> *Two points are addressed here: First, the uncertainty discussion is an average for a larger data set, which does not apply for individual profiles, and second, other processes add to the measurement uncertainty that are not touched upon here.*

> *We have slightly tweaked these two paragraphs to make this more clear:*

> *"The uncertainties discussed here describe the mean removal of systematic biases due to the time response of ECCs for the entire data set and help quantifying the uncertainty of ozonesonde profiles in the validation of remote sensing observations. Estimating the uncertainty of the correction of individual profiles, which depends strongly on the structure of the each profile, requires a more detailed analysis based on that profile structure.*

> *The corrections and uncertainties discussed here apply only to the time response model described above. Other effects, such as response differences of sondes from different manufacturers and pump related effects are not captured by the processes described here. However, the corrections for time response of the ECC need to be considered in properly quantifying other processes influencing the accuracy of ECC ozonesondes."*

Lines 544 to 548: This has been known and sometimes corrected since the 1980s. Maybe add one or two of these old references?

> *We added a paragraph about previous work on response time correction in the introduction:*

> *'The time response of the ECC has been studied in the past. De Muer and Malcorps (1984) studied the frequency response of Brewer Mast type electrochemical ozone sondes, which is based fundamentally on a similar chemistry as the ECC. They recognized that a convolution of different frequency responses is required to correct the time response of that sonde type. Imai et al. (2013) et al. applied a correction for the fast reaction pathway for the validation of SMILES satellite observations. Huang et al. (2015) derived a different correction for the fast reaction pathway, which in effect is very similar to that applied by Imai. However, none of the previous studies*

*considered the time response of the slow and fast reaction pathway and their connection to the "background current", as well as the fact that these processes require the use of a proper pump efficiency correction to avoid a compensation of biases.'*

**Reply to comments by Prof. Fujiwara**

*We appreciate the thoughtful comments by Prof. Fujiwara. We have considered these in the clarification of the manuscript, which was also suggested by the reviewers. We included short discussion of the time lag correction in the introduction and a reference to the earlier work by Imai et al. (2013), which we had missed. We also made the Figure legends and symbols more consistent throughout.*

*We would like to point out that the algorithm considers the temporal response of two separate reaction pathways. A slow reaction pathway, of which the instantaneous contribution has to be estimated, and a fast reaction pathway, for which a time lag corrected equilibrium value has to be calculated. The time lag correction is similar to what has been done before, including the work by Imai and coworkers. Considering the time response of both pathways simultaneously and their combined effect on pump efficiency and the historic "background current" is our new result.*

*We hope that the improved explanations and mathematical derivations clarify our main points.*

---

## Referee Report (RR1)

[referee-annotated manuscript omitted]

---

## Author Response (AR2)

**Replies to reviewer's technical suggestions**

We corrected almost all technical points the reviewer identified in our final manuscript. We did not follow the following technical points:

We did not specify a range for the excess electrons in the summary as suggested by the reviewer. This is one parameter that requires better quantification and we did not want to imply that we know the exact range.

The legend and caption in Figure 11 are correct and consistent with the colors used Figure 9.